# Pathomechanics of Early-Stage Lumbar Intervertebral Disc Degradation Leading to Discogenic Pain—A Narrative Review

**DOI:** 10.3390/bioengineering12040389

**Published:** 2025-04-05

**Authors:** Thomas Hedman, Adam Rogers

**Affiliations:** 1F. Joseph Halcomb III, M.D. Department of Biomedical Engineering, University of Kentucky, Lexington, KY 40513, USA; 2Spinal Simplicity LLC, Overland Park, KS 66211, USA; arogers@spinalsimplicity.com

**Keywords:** biomechanics, lumbar intervertebral disc, degradation, discogenic low back pain, pathoetiology

## Abstract

Although the existence of highly prevalent pain, disability, and work time lost associated with discogenic low back pain is well known, the recognition of the culpability of universally present disc degradation and mechanical insufficiency in the first three decades of life is often overlooked. There is a corresponding “treatment gap” and no current interventions with demonstrated capabilities to address the pain and resist the usual progression of increasing structural failure of spinal tissues with increasing levels of pain and disability. This narrative review summarizes more than forty years of the literature describing the pathomechanics of progressive degradation of lumbar discs, with a focus on studies that implicate an increasing mechanical insufficiency in the etiology of early-stage chronic and recurrent discogenic low back pain. Topics highlighted in this review include the deleterious biological changes that begin soon after birth, stress intensification due to the loss of fluid phase load support, fatigue weakening and damage accumulation in non-regenerative tissue, disc tears, segmental instability, and the timeline for first incidence of chronic low back pain. The review concludes with preferred treatment characteristics and a brief summary of emerging treatment approaches.

## 1. Introduction

There are several emerging intradiscal treatment strategies intended to address the widespread and inadequately addressed clinical problem of chronic or recurrent discogenic low back pain. While the goal of an early-stage treatment that is capable of stemming the cascade of progressive tissue damage, pain, and disability has been a “holy grail pursuit” for several decades, the path forward for these emerging treatments and the basis for predicting ultimate success remains unclear. Understanding the characteristics of progressive tissue degradation, especially the degradation that accumulates prior to the typical onset of pain and disability, is critically important. While there are multiple factors that contribute to early and progressive tissue deterioration in intervertebral discs, including genetics, nutritional insufficiency, and inadequate biological repair, the incremental loss of structural integrity and corresponding loss of mechanical capabilities leading to additional damage from mechanical overload may be the predominant factor resulting in early-stage discogenic pain. The vicious cycle of accumulating tissue damage due to overloading occurs in a unique way in lumbar discs due to the inadequate tissue repair and regeneration capabilities in the largest avascular tissue in the body [1]. This narrative review is kept from becoming voluminous by restricting its focus to degradation occurring within this early stage.

Degenerative factors such as deleterious mechanotransduction (i.e., cells converting aberrant mechanical stimuli to electrochemical activity that leads to upregulation of catabolic cellular functions), oxidative stress degradation, enzymatic degradation, and inflammation-related degeneration of discs can also result from the cascading mechanical overload of the tissue. The role of inflammation in disc degeneration remains controversial, as its role in maintaining tissue homeostasis and contributing to tissue regeneration is crucial [2]. While the impact of oxidative stress and enzymatic- and inflammation-related degradation may rise in significance with age, there is no evidence that they are dominant effects in younger discs. Likewise, it is reasonable to assume that a transition in the balance of anabolic to catabolic gene expression that results in an increase in matrix-degrading enzymes would be preceded by a significant degree and duration of mechanical degradation. There is an increasing yet incomplete understanding regarding the pathways of mechanotransduction, including the complex overlap of responses to specific stimuli and differences based on cell types and ages [3,4,5,6]. The timelines for these effects in patients are not known but are generally characterized as related to more advanced (compared to early-stage) degeneration and aging [3,4,5,7].

The impact of genetic factors other than genetically determined anatomical differences on lumbar disc degradation has been suspected since the seminal study by Battié et al. in 1995, where disc degeneration was assessed, and environmental and hereditary risk factors were compared in exposure-discordant monozygotic twins [8]. Within a few years of the Battié et al. paper, studies identifying specific genes associated with disc degeneration and disc bulge began to emerge [9,10]. While not the focus of this review of early-stage disc pathomechanics, it appears that at least one gene has been found in a large-scale study to have a strong association with disc degeneration in individuals younger than 40 years of age [11]. The influences genetic factors and aberrant gene expression [12,13] have on increasing the incidence of the early-stage vicious cycle of cascading mechanical degradation have not been determined. Noting the advent of technological advances in genetic therapies [14], it is still somewhat unclear whether knowledge of contributing genetic factors for early-stage discogenic back pain will play a role in treatment choices for individuals.

Pathomechanics refers to mechanical environment-induced pathogenesis, the mechanics of degraded or damaged mechanically functioning tissue leading to dysfunction and clinical symptoms. The timing and extent of this loss of intrinsic mechanical capabilities of a tissue, whose primary roles are load support and motion constraint, should influence the priority of this factor in the early stages of symptomatic disc degradation. Likewise, the correlation of mechanical insufficiency (the degraded tissue being unable to support loads and motions of daily life without incurring additional damage) with known mechanisms of pain generation suggests that addressing the declining tissue mechanical capabilities needs to be a priority in the early-stage treatment of this multifactorial disorder.

The intent of this narrative review is to provide an orderly overview of the pathomechanics at the core of early-stage intervertebral disc tissue degradation and the associated discogenic low back pain. The most widely used treatments for early-stage discogenic low back pain have not evolved over the past few decades. Clinical interventions continue to be primarily palliative in nature, reducing pain without addressing the core mechanical deficiencies until much later in the spectrum of this progressive disorder. Prevention of disease progression nearer the onset of the disorder has not been demonstrated to be an achievable goal and, therefore, the requirements for successfully achieving this lofty goal are ill-defined with a conflation of later-stage issues with the early-stage issues. In an attempt to produce some clarity, the current paper pulls together and highlights studies focused on the pathomechanics of lumbar intervertebral discs with a secondary purpose of aiding in the evaluation of emerging disc repair strategies for early-stage disease.

## 2. Methods

This narrative review primarily draws upon 40 seminal articles published between 1975 and 2010 on the topics of mechanical degradation of intervertebral discs and early-stage discogenic low back pain. More recent articles that further evaluated the findings of these influential articles were also reviewed. A foundational article focus was preferred to fit the theme of this narrative review, an organized summary of several core disc pathomechanics phenomena that would be useful considerations for emerging treatments of early-stage discogenic back pain. A secondary review was made of articles in the past 5 years (2020–2025) directed at emerging intradiscal therapies for discogenic back pain. In order to focus on the early stages of this progressive disorder, papers with a primary focus on later-stage disc degeneration or degenerative factors that stem from mechanical environment degradation were excluded from this report.

Article selection involved targeted searches for original articles with English full texts available on PubMed and Google Scholar using the following search terms: intervertebral disc and low back pain and mechanical degradation and (accumulated damage or fatigue damage or instability or annulus tears). Books, theses, and articles focusing on device or therapeutic approaches, methods of assessing joint instability, and animal models were excluded. To address the early-stage disease timeframe of this review, another similar search involved the search terms (chronic or recurrent) and low back pain and first incidence. Of the more than 400 articles identified in these searches, referenced articles were selected based on abstract screening, followed by full article reviews for relevance to early-stage discogenic back pain and related pathomechanics. Representative articles from often-cited authors were selected when multiple articles presented similar data or reviews.

In addition, terms such as (genetic factors and disc degeneration), (inflammation and disc degeneration), and ((mechanobiology or mechanotransduction) and disc degeneration) were manually searched in the above-mentioned databases for review articles and cited seminal articles in reputable journals. Screening of these articles was based on full reviews for content describing the likelihood of these mechanisms substantially contributing to early-stage disc disease.

## 3. Results

### 3.1. Early-Stage Changes in Biology and Biochemistry Affect Disc Biomechanics

In the late 1970s, Kirkaldy-Willis et al. [15] identified intervertebral disc degradation as the precipitating first phase of the progressive degenerative cascade in the lumbar spine. While the progressive nature of this disorder is clearly evident via medical imaging in mid- to late-stage degenerative cases, Buckwalter’s seminal article in the mid-1990s [16] highlighted the fact that deleterious changes in disc tissue begin soon after birth, with disc regenerative capabilities decreasing and disc tissue degradation increasing throughout the subsequent decades of life. Boos et al. [17] conducted a comprehensive histologic evaluation of discs obtained from deceased individuals, fetal to 88 years of age, in which they detailed the increasing levels of disc tissue breakdown by the first half of the second life decade (i.e., during adolescence) and the emergence of rim lesions in the second half of this decade (17 to 20 years of age). This early progressive degradation of lumbar discs was further explained by Urban et al. [18], who noted the deficient nutrient support, declining cell viability, and degradation and loss of water imbibing proteoglycans inherent in lumbar intervertebral discs. Loss of proteoglycans in both the nucleus and annulus regions of the disc results in a loss of fluid phase load support and a corresponding increase in solid-phase loading with general and regional intensification of tissue stresses [19,20,21]. Buckwalter [16] also noted the buildup of degraded macromolecules due to the inefficient outflow of waste products from the disc. The combination of a lack of repair capabilities and high and increasing tissue stresses contribute to progressive degradation of the normal collagen fibril organization and the accumulation of degraded matrix macromolecules [16,18,21].

### 3.2. Fatigue and Accumulated Microdamage

In 1988, Brinckmann et al. [22] demonstrated that *mechanical fatigue*—“the apparent strength of the material under repeated loadings is reduced with respect to the strength measured under a single load cycle”—was an important consideration in the degradation and failure of intervertebral joint tissues. They recognized that the inherent “repair time” of the tissue determined whether repetitive loading would cause microdamage to accumulate in the tissue, rendering it mechanically degraded. The focus of their paper was on the mechanical fatigue of vertebral bodies, not on the repair-deficient intervertebral disc. Adams and Roughley [21] concluded that tissue remodeling is virtually absent in the intervertebral disc annulus: “Collagen turnover time in articular cartilage is approximately 100 years and could be even longer in the disc”. So, unlike other load-supporting tissues, the fatigue-weakening effects of repetitive loading can be expected to add to the disc’s mechanical insufficiency and the resulting tissue microdamage, which is known to accumulate in these tissues. Microdamage accumulation, combined with stress increases from the loss of fluid phase load support, results in an increasing mechanical insufficiency of disc annulus tissues, which should be understood to be the norm, varying only by degree.

### 3.3. Disc Delamination and Tears: Early-Stage Prevalence and Clinical Relevance

Even at the early stages of degeneration, the structural changes to the annulus will include internal disc disruption, such as radial fissures and circumferential tears. Annular tears are found in 50% of lumbar discs under 35 years of age [23] and 100% of L4–L5 discs in the 10–30 year age group [24]. Circumferential tears, which are comprised primarily of delamination of adjacent lamellae [25], are the most common, can occur strictly from cyclic loading [26], and are the first to appear [23,24,25]. Schwarzer [27] found that 39% of low back pain patients had internal disc disruptions observed on CT discograms. Other early structural changes include inward buckling of the inner annulus, increased radial bulging of the outer annulus, reduced disc height, rim lesions, and endplate defects [28]. Figure 1 summarizes the factors known to contribute to early-stage lumbar disc degradation and the corresponding loss of load support and motion constraint.

It is noteworthy that while some disc tears and the initial loss of fluid phase load support may be detectable in vivo using MRI, accumulated microdamage, annular delamination and small tears, tissue compositional changes, and fatigue weakening of disc tissue cannot be detected, much less quantified, using current imaging technologies, including dynamic imaging techniques. While undetectable, these changes may be clinically consequential, producing disabling low back pain with no evidence of nerve root compromise [29]. While the prevalence of internal disc disruptions is exceedingly high, they may or may not cause any pain. Yet, considering the long-term disability associated with this progressive disorder, early and aggressive treatment at the early stages of the disorder would be preferred over delayed treatment, waiting until the disc disruption has progressed to a more detectable status.

### 3.4. Segmental Degenerative Instability

Because of the biomechanical role of the lumbar annulus fibrosus, accumulated microdamage of the annulus will directly affect disc mechanical function, leading to permanent tissue deformation (plastic deformation), radial bulging under load, and a reduction in passive tissue constraint of joint motion [26]. Momentarily setting aside the increased disc bulging under normal physiological loading associated with mechanically degraded discs, segmental instability is a focal point in the management of low back pain [30]. To fully appreciate the relationship between progressive disc degradation and segmental instability, it is best to look at instability as a progressive three-dimensional motion and deformation disorder that is not limited to a classical, dichotomous, translational, or rotational motion that surpasses a clinically determined limit. Clinical translational or rotational indicators of segmental instability are useful for standardizing and simplifying clinical decision-making. However, behind these clinically useful limits, segmental degenerative instability is the degradation-related progressive motion disorder where tissue mechanical degradation causes a gradual loss of segmental constraint and load support. Like core tissue degradation, degenerative segmental instability begins in the early decades of life and progresses thereafter. Loss of segmental constraint results in aberrant motions and deformations, which could, in turn, elicit discogenic pain. A common metric used to assess joint instability is the neutral zone [31]. This characteristic has been directly correlated with the incidence of low back pain [32]. As one of many possible examples of the progressive nature of this disorder, degenerative segmental instability could entail a gradual increase in neutral zone size due to a progressive reduction in passive joint constraint (Figure 2). Another example would include tissue mechanical insufficiency reducing joint constraint, which gradually increases translational slip when bending, eventually becoming a fixed anterior translation or spondylolisthesis. Another example would be the gradual increase in joint angulation under flexion loading due to a reduction of stiffness and load support in the degraded disc. Therefore, the loss of lumbar intervertebral joint constraint due to the degradation of annulus tissues can be mechanistically linked to discogenic pain. The point at which this progressive segmental instability begins to contribute to episodes of low back pain, if that occurs, is likely to be different from individual to individual.

Whether by acute events, genetic factors [11], or a result of the progressive disc degradation described here, increased disc bulge due to insufficiency of the degraded disc has been shown to have a bulge magnitude [33] that is three times the displacement required to elicit a neural response in mechano-nociceptive nerve fibers of the type found in the outer annulus of the intervertebral disc [34,35,36,37]. Consequently, increased disc bulge resulting from loss of axial compressive load support due to degradation of annulus tissues can be mechanistically linked to discogenic pain.

### 3.5. Pathomechanics and Low Back Pain

#### 3.5.1. Discogenic Pain

Because this growing mechanical insufficiency begins early in life, and the disc annulus is the primary anatomical structure providing load support and motion constraint in intervertebral joints [38], it follows that the early degradation of annulus tissue can be a primary contributor to the etiology of lumbar joint tissue failures and pain, both early-stage and later-stage. Despite the primary role of the annulus in lumbar joint mechanics, the majority of historical and emerging intradiscal treatment approaches have been directed to the nucleus pulposus region of the disc. The reasons for this are not clear but may lie in an incorrect assumption that nucleus degradation precedes annulus mechanical degradation. This assumption may be, in part, due to degradation detectability. Nucleus changes can be more readily observed by magnetic resonance imaging due to the loss of water content compared to changes in the mechanical integrity of the fibrous, load-carrying annulus.

It is suggested in this paper that discogenic low back pain experienced in the third and fourth decades of life (20s and 30s) can often be attributed to the accumulated damage to lumbar discs in the first 3 decades of life and the resulting disc pathomechanics associated with this degradation. Psychosocial factors play a role in the chronicity of low back pain, but inadequate biological repair and progressive mechanical insufficiency are the sparks that light the fire. Lumbar disc tissue changes are universally present in the first three decades of life; however, they are not always associated with symptoms. While all repair-limited and mechanically degraded discs are not necessarily symptomatic, all persistent low back pain cases have in common disc degradation and the associated mechanical insufficiency of this primary load-carrying and motion-constraining tissue [39,40].

Since the early 1990s, tissue degradation and aberrant mechanical stress have been linked to low back pain incidences in several studies. The simple fact that mechanical provocation of discs can reproduce severe and chronic back pain indicates the role that mechanical stress and, therefore, tissue mechanical insufficiency can play in pain generation [27,41,42,43]. The outer annulus fibrosus is known to contain abundant nerve endings [44] that may be responsive to aberrant strains, deformations, and motions.

#### 3.5.2. Peak Age of Onset of Low Back Pain

Consistent with the evidence implicating early-stage mechanical degradation of lumbar discs in the etiology of recurrent and chronic low back pain, several studies demonstrate that the pain and disability associated with this degradation typically begins at the early stages of a person’s adult life and progresses for decades thereafter [45] during the primary working years of the individuals. Deyo and Tsui-Wu, in 1987 [46], found the peak age of onset of low back pain to be between 20 and 29 years, with new cases declining after age 29. Likewise, Laslett et al., in 1991 [47], found that nearly 50% of New Zealanders with low back pain suffered their first episode before the age of 30 years. Similarly, Biering-Sorensen, in 1983 [48], when evaluating 1-year follow-up questionnaires of 30-, 40-, 50-, and 60-year-olds, found the incidence of first attacks of low back pain was highest in the 30-year-olds and decreased in the older age groups.

#### 3.5.3. Prevalence of Low Back Pain

For a variety of reasons, this debilitating disorder is widespread in Western society. Over 80% of the population experiences an episode of LBP at some point in their lifetime [49], with lifetime recurrence rates up to 85% [50,51]. The point prevalence in the adult population of degenerative disc disease related pain is 37%, and the 1-year prevalence is 76% [52], with more than two-thirds of recent onset cases and four-fifths of non-recent onset cases being repetitive (episodic and lasting more than a year) or chronic [40,45,53].

Chronic low back pain (CLBP) is more prevalent in women and increases linearly from the third decade until about age 60 [54,55], with 10–12% of the population disabled by CLBP [19,53]. As a consequence, degenerative disc-associated low back pain is the second most common pain condition, resulting in lost work time (first is headache) [56,57], with a total annual cost in the US estimated to exceed 100 billion dollars [58].

#### 3.5.4. Acute Back Pain

Loss of passive motion constraint can increase the demand placed on active joint constraint systems, in particular the multifidus and other components of the lumbar musculotendinous systems. Musculotendinous strains and sprains comprise a small proportion of acute back pain [45], typically resolving in a matter of weeks. The degree to which this acute back pain is, at least, in part, due to deficient passive load support and motion constraint is not clear. A reasonable assumption would be that reduced passive motion constraint and load support due to the mechanical degradation of the disc shifts some of the burdens of motion constraint and load support to the musculotendinous structures, increasing the possibility of muscle and tendon strains and sprains, leading to acute but recurrent inflammation and pain. The fact that the majority of episodic back pain is recurrent [40,45,49] implicates underlying deficiencies in passive load support and motion constraint systems in the pathoetiology of this disorder. The involvement of mechanically deficient passive systems in recurrent overloading of active systems is not surprising, considering that all passive constraint tissues have experienced degradation by the third decade of life.

### 3.6. Preferred Treatment Characteristics for Early-Stage Discogenic Back Pain

With the advent of more refined imaging techniques and methods of diagnosing discogenic low back pain, the diagnosis of “discogenic, recurrent low back pain” is beginning to replace the previously common identifier of “non-specific low back pain” [29]. Arguably, the term “non-specific low back pain” wrongly suggests that a likely reason for the pain cannot be determined when the universal and not insignificant degree of lumbar disc mechanical degradation is well established, as described in this review. The fact that disc mechanical degradation is not always associated with pain does not take away from the associations of progressive mechanical degradation with aberrant motions and deformations known to be causative in discogenic pain through known pathways.

Considering that early-stage discogenic pain and disability stem from core mechanical deficiencies, appropriate interventions will be directed to ameliorate the mechanical degradation both to address current pain and disability and with the intent to intercept the progressive cascade of degradation, leading to increasing tissue degradation and worsening symptoms (Table 1). Therefore, a successful intervention will act to restabilize the affected segment and add load support where it has been lost to fatigue weakening, loss of fluid phase load support, and accumulated microdamage. Preferably, this added motion constraint and load support will not alter or interfere with the mechanical loading patterns inherent in the disc tissues, as this could potentially contribute to additional degradation and loss of the mechanical attributes of the tissues. Ideally, the added mechanical support and constraint are immediately effective rather than requiring the passage of several weeks or the cumulative effects of multiple administrations.

Palliative treatments, including epidural steroid injections, opioid and non-opioid analgesics, and nerve ablations, are perhaps contraindicated in early-stage disc degradation-related pain and disability cases. Pain masking could be detrimental to already mechanically insufficient tissues due to a corresponding absence of pain-avoidance constraints on movements while undergoing daily activities, leading to a greater frequency of deleterious loading events.

Demonstrations of joint re-stabilization and augmentation of disc mechanical load support are, therefore, important indicators of a potentially appropriate intervention. Considering the progressive nature of this disorder, the ability to restabilize degraded lumbar joints and increase disc annulus strength and other mechanical properties following the intervention are perhaps as important as demonstrations of rapid and long-term reductions of pain and disability. To that end, mechanical benefits in both tissue strengthening and motion constraint can be most clearly evaluated and quantified by in vitro cadaveric tissue studies or from animal ex vivo experiments for interventions that rely on biological response. The limitations inherent in animal models with regard to representing degenerative human discs will equally limit the value of the results from investigations of the mechanical benefits of biological therapies. Mechanical benefits demonstrated in cadaver experiments or from tissues harvested following in vivo animal administration would ultimately need to be confirmed in human clinical studies, with the understanding that confirmation of mechanical changes resulting directly from an intervention may have to rely exclusively on changes in patient kinematics following the treatment compared to baseline characteristics.

Protection of disc tissues and adjacent musculoskeletal tissues by the use of non-destructive microinvasive methods is essential for an intervention at the early stages of this chronic and progressive disorder. Preservation of tissue anatomy and avoidance of treatment adverse effects that could potentially be detrimental to the disc or adjacent tissues are vitally important at this early stage. The common motto is to “not burn any bridges”. With tissues already showing symptomatic mechanical insufficiency, the intervention should be strictly beneficial mechanically, especially in the essentially avascular intervertebral disc tissues, which have very limited capacity for healing or regeneration.

The durability of the treatment effect is another important consideration in the success of interventions for early-stage disc degradation-related pain and disability. Permanent, non-degrading mechanical support for the degraded disc is the ideal remedy. Patient compliance is typically at odds with the need for repeated treatments; therefore, the less frequently that a restabilizing treatment would have to be administered to be effective, the better.

Healthcare and societal cost considerations are also paramount. Expensive interventions face an uphill battle for reimbursement by the insurance industry and those making decisions regarding public healthcare cost coverages. Cost benefits for treatments that reduce the likelihood of follow-on procedures and expensive surgeries may only be fully calculable after long-term cost and effectiveness studies. However, it is important to note that early-stage and durable treatments that reduce the cost of care (relative to traditional joint fusions, decompression surgeries, and disc replacement surgeries) or the need for ongoing visits to a healthcare provider could provide a vital treatment option to marginalized and economically challenged groups, as well as provide a global treatment option for those countries or regions that do not have ready access to more invasive solutions. Another consideration is the accessibility of the treatment to physicians in terms of procedure simplicity, minimal learning curve, and not requiring specialized or advanced equipment.

Lorio et al. [59] describe intradiscal interventions as filling “the extensive treatment gap between conservative management and traditional spine surgery”. Considering the early onset of recurrent discogenic pain described above, this treatment gap often involves one to four decades of pain, disability, and loss of workdays prior to patients and providers accepting the suitability of expensive and higher-risk spinal surgery.

Even after new technology has demonstrated the ability to address early-stage disc degradation-related pain and the ability to resist the progression of degradation of the disc and surrounding load-supporting tissues, improved clinical care for this widespread disorder will require adoption by all the medical care stakeholders, including frontline clinicians (primarily general practitioners), spine interventionalists and surgeons, clinical practices and hospitals, and medical insurance providers. Beyond filling a treatment gap in the current continuum of care, a successful early-stage discogenic back pain intervention would potentially reduce the need for traditional spine surgery at the later stages of the treatment continuum. With one component of potential healthcare cost savings coming from the prevention of progressive lumbar tissue deterioration, cost advantages will only be fully demonstrated with long-term and large clinical studies with real-world patients to quantify the reduction of post-intervention costs associated with repeated visits to healthcare providers, increasing severity of the disorder (progression to disc herniation, radicular pain, spinal stenosis, spondylolisthesis, etc.), and avoidance of expensive surgeries. In addition to these follow-on healthcare costs, the cost reduction of fewer workdays lost to recurrent back pain during what would otherwise be a treatment gap period should also be carefully evaluated. One medical insurance-related hindrance to obtaining the data to propel a significant change in patient care, a “catch-22” of sorts, is that novel treatments typically have to demonstrate cost savings and effectiveness before long-term and large studies of this type are financially feasible.

With the emergence of intradiscal interventions that are capable of re-stabilization and mechanical load support of degraded lumbar discs, it is expected that there will need to be a paradigm shift in the continuum of care for this disorder, from a reliance on conservative care followed by “watchful waiting”, physical therapy, nerve blocks, and other palliative care standard treatments, placing the new therapeutic as the preferred approach at the early stage of treatments. Until recently, virtually all treatment guidelines for the management of chronic “nonspecific low back pain” could not recommend intradiscal injections until supporting, high-quality, randomized controlled trial data were available [60]. Intradiscal treatments are increasingly used today for recurrent and chronic discogenic low back pain, but spine health societies and payers are still reticent to provide their full recommendation for any of these procedures. Quality randomized controlled trial data are indeed an appropriate standard. Notably, the current standard treatments (or non-treatment) set a low bar for treatment superiority when patient-reported outcomes, costs and frequency of repeated physician visits and follow-on procedures, use of analgesic medicines, and reducing work time lost are all included in the assessment.

### 3.7. Emerging Intradiscal Therapies for Early-Stage Discogenic Pain

The recent paper by Lorio et al. (2024) [59] is referenced here as a relatively comprehensive listing of the current state of the art regarding intradiscal injections in the treatment of discogenic pain. Most of the candidate therapies listed in this paper are biologic in nature: mesenchymal stromal cells (MSCs), platelet-rich plasma (PRP), nucleus pulposus structural allograft, and other cell-based compositions. Other emerging intradiscal approaches referenced in Lorio et al. include Discseel, which uses allogeneic fibrin; Hydrafil, a polymer composite hydrogel augmentation material; and Discure, a multi-electrode implanted catheter that provides intradiscal electrical stimulation in an attempt to increase the osmotic gradient and induce increased hydration in the degraded disc. Another emerging intradiscal approach is Intralink, an injectable, self-polymerizing nano-tether mesh that uses a self-forming polymeric mesh that covalently bonds to collagen in the annulus.

#### 3.7.1. Mechanisms of Load Support in Emerging Intradiscal Therapies

It can be presumed that biological therapies employ two general mechanisms for augmented load support and motion constraint: (1) regeneration or repair of the load-supporting tissue and (2) increased nucleus hydration producing increased liquid-phase load support, theoretically reducing stress on the solid-phase disc components (Table 2). In the case of nucleus allograft implants, increased nucleus volume and hydration without repair of the degraded annulus may heighten mechanical stress in the annular fibers, leading to accelerated degradation of already-degraded annular tissues rather than resisting the progression of annular degradation. Lying behind this goal of increasing nucleus volume, there may be a sense that just as a flat tire needs to be reinflated to support a vehicle’s weight and function properly, addressing the loss of nucleus volume may enable the disc to function properly, including re-establishing disc height to reduce compression of surrounding nerve tissues. Using the same analogy, reinflating a degraded and worn-out tire may be short-lived, or it may lead to a more dramatic failure of the tire if the mechanical deficiencies of the solid portions of the tire are not also addressed.

The allogeneic fibrin sealant is not capable of providing load support or motion constraint without the addition of the Factor XIII-A crosslinking component.

After it cools to body temperature, the polymer composite hydrogel acts as a bulk filler, a compressive load-carrying intradiscal solid material. It may be important to note the contrast between the mechanisms of load support between the disc annulus collagen matrix and the bulk filler. Collagen molecules and fibers in the disc annulus resist tensile stresses, while a bulk filler resists compressive forces. It is not known how these contrasting mechanisms of load support by adjacent materials (native tissue and implant material) may affect the long-term integrity of the disc tissues.

The multi-electrode catheter increases the osmotic gradient in the disc to induce increased hydration, potentially leading to annular regeneration [61] in the degraded disc. This electrical stimulation method of increasing disc hydration will face the same or greater challenges with regard to the regeneration of load-supporting tissues as biological therapies. Likewise, similar to nucleus allograft implants, increased nucleus volume and hydration without repair of the degraded annulus can be expected to increase mechanical stress in annulus fibers, leading to accelerated degradation of annular tissues rather than resisting the progression of annular degradation.

Unlike the compressive load support of bulk fillers in the disc, the intra-annular injectable polymer mesh technology adds a mesh of tensile load-carrying polymers that covalently attach directly to the collagen and help to support the degraded, tensile load-carrying fibrous collagen in the annulus. In this way, the polymeric mesh adds tensile load support to the tensile load-resisting collagen network of the annulus, increasing joint constraint in a mechanically similar way to the native tissue fibers.

#### 3.7.2. Comparison of Emerging Intradiscal Therapies

The regenerated or repaired disc tissue is a worthy goal of biological therapies, and these therapies have been successfully developed for tissue repair in other load-supporting tissue applications. However, the avascular disc has an inadequate nutritional supply to support added cells or biological activity. The harsh biological conditions that led to the pre-existing degradation pose challenges to the durability of any desired regenerative effect.

Lorio et al. [59] describe the “overriding challenge” for biological agents to provide a durable restorative effect lies in the harsh biological conditions of the lumbar intervertebral disc. In addition to the challenges of declining cell viability and loss of proteoglycans discussed earlier in this review, the aging disc is a poor environment for biological repair or regeneration due to thickening and calcification of the cartilaginous endplates, the primary nutrient inflow pathway to the disc [18,20]. This disrupted nutritional pathway leads to low oxygen and glucose concentrations and acidic pH from higher lactic acid concentrations. The native cells succumb to these harsh conditions, with more than half of adult disc cells appearing necrotic or apoptotic [20]. One might question whether it is a rational choice to attempt to employ a cell-based or other biological therapeutic in a nutritionally deficient tissue [18] that has been declining in cell viability since birth [16]. After more than two decades and immense worldwide research investment in both funding and human capital, the sage advice provided by Urban et al. [18] continues to ring true: “New methods of disc repair involving stimulation of native cells or insertion of new cells or tissue-engineered disc should, however, be used with caution in humans. For successful disc repair, the newly inserted or stimulated cells have to exist in conditions where they remain viable and active. It is thus essential that the nutrient supply to the disc is adequate and, moreover, that it can support the increased nutritional demands these methods induce; the disc will need more nutrients to support inserted cells or cells for which the activity has been increased”. Urban et al. continues, “it is unrealistic to expect that reimplantation of cells into such discs would effect a repair. Before cell-based therapies can be introduced successfully, some method of selecting suitable patients on the basis of an adequate nutrient supply to the affected disc appears absolutely crucial”.

Similarly, Lorio et al., some twenty years later, describe the requirements for durable clinical success for biological agents, including the ability to remain viable in the degenerated disc long enough to contribute to matrix production, the ability to generate adequate paracrine signaling to alter the behavior of native cells, support the recruitment of regenerative cell types or limit infiltration of fibrotic/catabolic cells, and provide intradiscal mechanical support, aligning with the theme of this article.

Other disadvantages of biological approaches include clinical success often turning out to be patient-specific with distinctly different outcomes for “responders” and “non-responders”, the treatments can be costly or require more than one procedure to initiate the repair, and biological treatments by their nature generally require several weeks to demonstrate an effect.

It is well known that fibrin is one component of blood clots, but fibrin clots are unstable and break down without the crosslinking component transglutaminase, an enzyme that catalyzes the crosslinking of proteins by forming covalent bonds between lysine and glutamine residues in various polypeptides. Factor XIII-A is the active transglutaminase that plays a crucial role in the coagulation cascade. While tissue sealing effects, including forming a physical barrier to resist migration of catabolic agents, may occur in the short term after intradiscal delivery of fibrin, it is debatable that these effects would endure long enough to be consequential clinically without adequate transglutaminase-catalyzed covalent crosslinking to make this sealant stable and more resistant to breakdown. Longer-term effects, such as resisting neoinnervation into the degraded annular fibers, are unlikely to occur without a stable sealant construct, which fibrin alone does not provide.

The Hydrafil composite hydrogel augmentation material (PVA/PEG/PVP/ barium sulfate) attempts to remedy the mechanical insufficiency of the disc by providing compressive load-carrying implant material to the degraded disc. The hydrogel also has high hydrophilicity to draw in and retain interstitial water in the disc. Hydrafil has demonstrated improvements in back pain and disability in early studies, with no persistently symptomatic serious adverse events [62]. The hydrogel is injected as a relatively high-temperature liquid (65 °C) into the nucleus pulposus tissue using a relatively large needle (17Ga) with the intention to also flow into the annulus region of the disc. In order to not displace or disrupt disc tissues, these fillers require gaps within the disc, which is generally limited or not available in early- to mid-stage degenerated discs. While it may have been initially intended for early-stage disc degeneration cases, Hydrafil is directed toward more severely degenerated discs (moderate to severe, Modified Pfirrmann Levels III-VI) that can accommodate the addition of a bulk material. Additional concerns include the potential for disc tissue damage in proximity to the implant material when it is injected as a hot liquid. Collagen can begin to denature at temperatures above 60 °C, and cell death is likely to occur at those temperatures. The depth of tissue damage and cell death from contacting the temporarily hot liquid is not known and may be insignificant. The relatively large diameter needles used in this procedure may lead to implant material expulsion, as well as cause “drastic alteration” of annulus strain behavior, leading to disc degeneration [63,64]. Initial clinical results showed relatively high rates of implant displacement or extrusion (15%) and serious adverse events (25%) [62] by 6 months.

Discure, an implanted multi-electrode catheter, provides intradiscal electrical stimulation to increase the osmotic gradient in the degraded disc [61]. The multi-electrode catheter is connected to an implanted pulse generator to provide electrical stimulation in the degraded disc. This approach is hindered, similar to biological therapies, due to the disc having a deficient nutritional supply and harsh biological conditions, making it less likely that a change in disc hydration would lead to a durable regenerative effect. However, a pre-clinical organ culture experiment using porcine intervertebral discs demonstrated several markers consistent with annular regeneration, including an increase in anti-inflammatory cytokines and a decrease in pro-inflammatory markers. Whether these effects can be consistently achieved with lasting effects in clinical settings with degraded and nutritionally deficient human discs has yet to be demonstrated.

Hedman et al. 2024 [65] recently discussed the capabilities of Intralink, an injectable, self-polymerizing nano-tether mesh, in the treatment of lumbar disc degradation and the associated pain and disability. The intra-annular mesh is comprised of a large number of genipin oligomers (relatively short monomers) that diffuse through the annulus tissue and covalently bond to amines on collagen. Intralink-treated cadaveric disc experiments have quantified the mechanical effects of the intra-annular polymeric mesh, including reduction of disc bulge under load, increasing resistance to annulus shear and delamination, increasing tissue tensile strength, and increasing intervertebral joint stability.

The results from these in vitro disc studies have begun to be replicated and confirmed in early clinical studies of the device [66], including demonstrations of immediate (1 to 2 weeks) and lasting (up to 2 years post-treatment) reductions of pain and disability coupled, with kinematic data showing an increase in segmental stability, especially in painful joints with segmental instability, assessed pre-treatment, which were more than 1.5 standard deviations above the asymptomatic mean. These data suggest that the intra-annular polymeric mesh can be expected to provide durable motion constraint and load support when added to the degraded disc annulus tissues and that addressing the core mechanical insufficiencies of degraded lumbar discs can result in immediate and lasting reductions of pain and disability. The early clinical data suggest an absence of serious adverse events in the use of this injectable treatment. The early clinical data also indicate that a second “booster” treatment may be required for high mechanical demand (heavier) patients. Additional clinical evidence from larger clinical studies and randomized controlled trials is needed to establish the stage of disc degradation that is best suited to be treated using this technology.

## 4. Discussion and Summary

All of the emerging intradiscal approaches discussed in this review have in common the intention to address the mechanical deficiencies of the degraded intervertebral disc. Because of the mechanical deficiency at the core of this debilitating pain, demonstrating the ability to improve clinical outcomes while also providing long-lasting load support and motion constraint is the key to having a successful intervention at the early stages of this progressive disorder.

If the current paradigm of clinical care for early-stage, recurrent, discogenic low back pain is to change, the restoration of lumbar joint mechanics and resistance to mechanical degradation are perhaps the most important aims for the advancement of care in order to leverage the support of patients, clinicians, and payers. Unfortunately, well-designed in vitro cadaveric studies (for devices) and ex vivo studies (for treatments that require a bodily response to fully achieve their intended purposes) that demonstrate the extent to which a treatment achieves its mechanical goals are sparse. Elimination of the current treatment gap will also rely on the ability of the emerging treatment to be delivered without disrupting spinal tissues and on a cost profile that is not prohibitive for widespread use.

This review is intended to highlight the need to address the core mechanical deficiencies of lumbar discs that appear to be only minimally degraded as a top priority in the management of early-stage discogenic low back pain. The impact of successfully addressing this ubiquitous mechanical degradation at the early stages of this progressive disorder is not known at this time. However, the immense healthcare and societal costs of this disabling malady have compelled a few innovators to seek a micro-invasive, intra-discal means of restoring the mechanical integrity of this ultra-important load-supporting tissue.

## Figures and Tables

**Figure 1 bioengineering-12-00389-f001:**
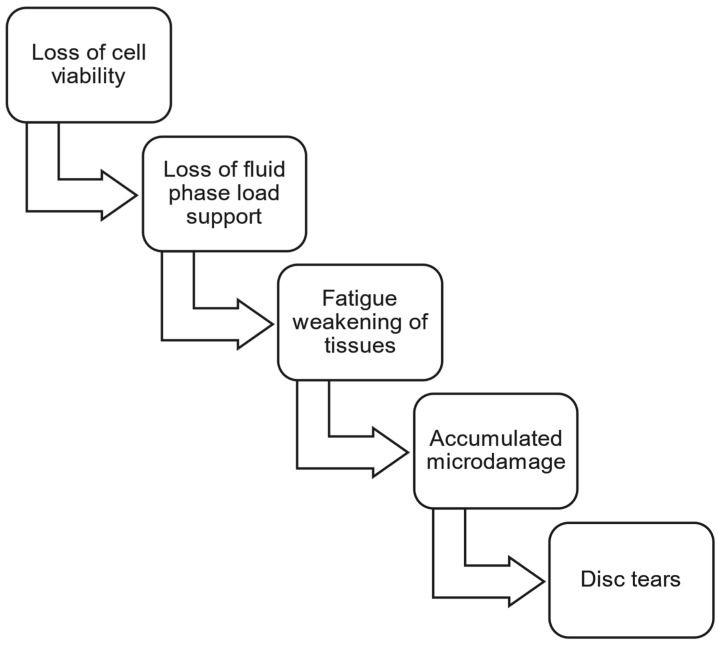
Factors contributing to the vicious cycle of disc structural degradation and mechanical insufficiency during the first 3 decades of life: loss of cell viability [15,18]; loss of fluid phase load support [19,20,21]; fatigue weakening of tissues [1,21,22]; accumulated microdamage [16,22]; and disc tears [23,24,25,26,27,28].

**Figure 2 bioengineering-12-00389-f002:**
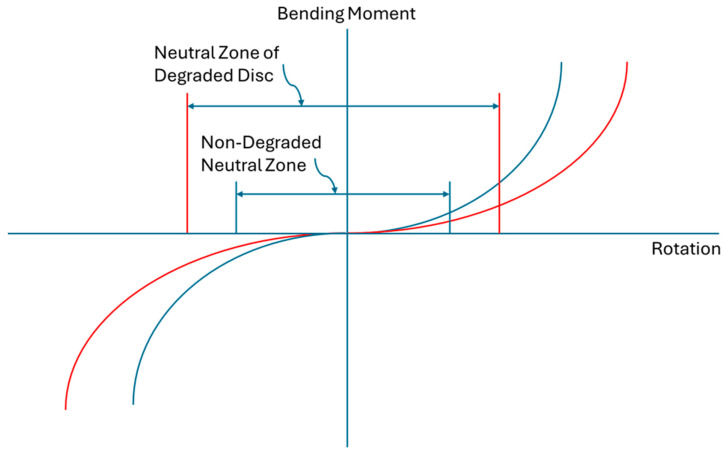
Schematic depiction of loss of lumbar intervertebral joint motion constraint resulting from progressive mechanical degradation of disc tissues. Non-degraded joint motion is depicted by the blue curve; degraded joint motion is depicted by the red curve. Increased neutral zone reflects a loss of passive tissue constraint and increased instability in the region of the motion curve that has the least resistance.

**Table 1 bioengineering-12-00389-t001:** Criteria (targets) for successful early-stage therapy for discogenic pain and disability.

Criteria	Rationale
Effective in reducing pain and disability	Primary reason patient sought medical attention
Provide load support and motion constraint, joint re-stabilization	To address cause of pain and to resist progression of disorder
Immediate effectiveness	Preferred by patients
Lasting effect, durability	To reduce repeat visits to healthcare providers, reduce cumulative healthcare costs
Single procedure	To avoid patient non-compliance with multiple procedure visits
Demonstrated mechanical benefits in the lab and in the clinic	To aid adoption of new therapy by demonstrating key benefit over palliative care
No adverse effects to disc or adjacent tissues	To promote long-term health benefits
Procedure simplicity, not requiring specialized equipment or training	To promote availability of treatment, remove potential hindrances to physicians
Healthcare and societal cost reductions	To encourage adoption by all the medical care stakeholders
Accessibility to marginalized and economically challenged groups, countries	Reducing treatment cost, time away from work, and need for repeated treatments helps the inclusion of these groups

**Table 2 bioengineering-12-00389-t002:** Mechanisms for increasing load support and motion constraint in emerging therapies.

Emerging Therapy	Load Support Mechanism (s)	Comments
Biological therapies—MSCs, PRP, allograft, tissue engineered materials, etc.	(1) New or repaired extracellular matrix to restore mechanics; (2) Restored disc hydration, liquid-phase load support	Regenerated or repaired tissue is a worthy goal, and biologic therapies work in other load-supporting tissues; however, the disc has deficient nutritional supply to added cells, challenges to durability of effect due to harsh biological conditions that led to pre-existing degradation, patient specific outcomes, and delayed effect
Nucleus allograft	Increased nucleus hydration, liquid-phase load support	Nucleus augmentation without repair of degraded annulus may increase stress in annulus and not resist progression of mechanical degradation
Allogeneic fibrin sealant	None without Factor XIII-A crosslinking component	Fibrin is unstable and breaks down without the crosslinking component
Multi-electrode implanted catheter that provides intradiscal electrical stimulation	Increase nucleus hydration, liquid-phase load support	Nucleus augmentation without repair of degraded annulus may increase stress in annulus and not resist progression of mechanical degradation
Polymer composite hydrophilic hydrogel augmentation material	Compressive load-supporting bulk filler material, increase fluid-phase load support	Disc augmentation without repair of degraded annulus may increase stress in annulus and not resist progression of mechanical degradation, space for filler may not be available in early-stage discs, potential tissue damage from hot injectate, possible implant material expulsion
Injectable, intra-annular polymer mesh	Mesh of tensile load-carrying polymers attached to collagen matrix	Polymeric mesh augments collagen tensile load-carrying to provide durable motion constraint and load support, requires second treatment for high mechanical demand patients

## Data Availability

No new data were created or analyzed in this study. Data sharing is not applicable to this review article.

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
