# Peer review of "Pathomechanics of Early-Stage Lumbar Intervertebral Disc Degradation Leading to Discogenic Pain—A Narrative Review"

_bioengineering, 2025, doi:10.3390/bioengineering12040389_

Round 1
Reviewer 1 Report
Comments and Suggestions for Authors
The introduction effectively outlines the clinical significance of discogenic low back pain and establishes the rationale for reviewing pathomechanics. However, it assumes prior familiarity with terms like pathomechanics and mechanical insufficiency, which may alienate non-specialist readers.
The review integrates foundational studies (e.g., Kirkaldy-Willis, Buckwalter, Boos) and recent work (e.g., Lorio et al., 2024; Hedman et al., 2024). However, there is a reliance on older seminal papers (1970s–2000s) and limited inclusion of recent advancements (2023–2025) in biological therapies or imaging techniques. Notably, while emerging treatments like Intralink and Hydrafil are discussed, newer studies on mechanotransduction, inflammation, or genetic factors in disc degeneration are underrepresented.
The narrative follows a logical progression (degradation biology → mechanical effects → pain → treatments), but subsections under Results (e.g., "Segmental Instability," "Existing Treatments") are overly dense. Subdividing these with more specific subheadings (e.g., "Disc Tears: Prevalence and Clinical Relevance," "Mechanisms of Load Support in Emerging Therapies") would improve readability.
Some details in Results (e.g., lengthy descriptions of Hydrafil’s mechanics) would fit better in the Discussion.
The Methods section is insufficiently detailed for replication. Key omissions include: Search terms (e.g., specific MeSH terms) and date ranges; Number of screened/included studies; Criteria for excluding "later-stage degeneration" studies; Risk of bias assessment tools or processes. Adding a PRISMA flowchart and transparent methodology would address these gaps.
The statement “Articles relevant to the topic were obtained from the first author’s journal article archive and searches in the PubMed database” (lines 44–46) is insufficient for the following reasons: 1) Lack of Transparency: No details are provided on search terms, date ranges, languages, or inclusion/exclusion criteria (e.g., study types, sample sizes). 2) Selection Bias: The reliance on the first author’s archive introduces an uncontrolled risk of cherry-picking studies that align with pre-existing views. 3) Reproducibility: Without explicit search strategies, readers cannot verify the comprehensiveness of the literature review or replicate the process.
Even narrative reviews must adhere to basic scholarly standards to avoid bias and ensure credibility. The omission of methodological specifics undermines the validity of the authors’ conclusions.
While the article implicitly resembles a narrative review, the authors need to explicitly state this in the title, abstract, or introduction.
Even narrative reviews must outline their foundational approach to literature selection. The authors omit: 1) Search Protocol: Specify databases beyond PubMed, search terms (e.g., "disc fatigue," "annulus degradation"), and filters (e.g., human studies, English-only). 2) Time Frame: Define the period covered (e.g., "1970–2024"). 3) Exclusion Criteria: Clarify why studies focused on late-stage degeneration were excluded (line 47). 4) Rationale for Source Selection: Justify reliance on the first author’s archive and potential conflicts of interest (e.g., the authors’ affiliation with Spinal Simplicity LLC, line 4–6). Without these details, the review risks being perceived as opinion-driven rather than evidence-based.
The methodological shortcomings reduce the review’s utility for guiding clinical decisions or evaluating emerging treatments (as claimed in lines 41–42). For example: The exclusion of studies on later-stage degeneration (line 47) is not quantified or justified, potentially omitting critical insights into early-stage progression. The emphasis on mechanical insufficiency (e.g., lines 85–89) relies heavily on selected historical studies, yet the lack of a systematic search raises questions about confirmation bias.
If Figure 2 is adapted, cite original work in the figure caption (e.g., "Adapted from [15]").
Reviewer 2 Report
Comments and Suggestions for Authors
Review of "Pathomechanics of Early-Stage Lumbar Intervertebral Disc Degradation Leading to Discogenic Pain - A Review" by Thomas Hedman and Adam Rogers
The manuscript addresses the pathomechanics of early-stage lumbar intervertebral disc (IVD) degradation and its link to discogenic pain. While the topic of the review is highly relevant, the reviewer finds the purpose and motivation behind this review are not clearly stated.
The reviewer identifies the following major concerns:
- The manuscript explores intervertebral disc (IVD) degradation and its association with discogenic pain, a topic that has been extensively covered in existing literature. To enhance its relevance and contribution, the authors should clearly distinguish their review from previous works by other research groups and explicitly articulate the novel insights or perspectives they bring.
- The manuscript is predominantly text-based, lacking original experimental data, comprehensive figures, or a quantitative synthesis of prior studies. The review contains only two figures and no tables, which is unusual for a review article.
- The organization of the manuscript needs improvement, as the flow of sections is difficult to follow, making it challenging to track the progression of ideas. To enhance readability and coherence, the authors should consider restructuring the content logically and including a clear table of contents at the beginning to provide an overview of the review.
At this stage, the reviewer considers a substantial revision necessary before the manuscript can be considered for publication.
Round 2
Reviewer 1 Report
Comments and Suggestions for Authors
The revised manuscript (v2) addresses several key points from the feedback on v1, but some gaps remain.
Original Feedback was: overreliance on older studies; limited 2023–2025 advancements. v2 added recent studies (e.g., Lorio et al. 2024, Hedman et al. 2024) and discussed emerging therapies (Intralink, Hydrafil) and biological advancements, but focus remains weighted toward foundational studies; newer mechanistic insights (e.g., inflammation, genetics) remain underrepresented.
Original feedback: Insufficient detail for replication. v2 added search terms (e.g., "intervertebral disc and low back pain and mechanical degradation"), databases (PubMed/Google Scholar), timeframes (1975–2010 foundational; 2020–2025 for emerging therapies), and clarified exclusion criteria (later-stage degeneration, animal/device studies). But, No PRISMA flowchart. It isn't so difficult, why not?
Original feedback: Reliance on first author’s archive introduced selection bias. No explicit justification for archive use; selection bias risk remains.
Expand rationale for excluding later-stage degeneration studies.
Strengthen discussion of biological therapies’ limitations (e.g., nutritional challenges).
Reviewer 2 Report
Comments and Suggestions for Authors
The authors have generally addressed the main points raised in the first review. While the review remains largely descriptive and narrative, the early-stage focus and the discussion of emerging therapies provide relevant information. The manuscript could be considered a useful contribution to the current state of the art.
Author Response
Comment 1: The authors have generally addressed the main points raised in the first review. While the review remains largely descriptive and narrative, the early-stage focus and the discussion of emerging therapies provide relevant information. The manuscript could be considered a useful contribution to the current state of the art.
Response 1: Thank you for your comments and helpful review of this manuscript.